# Effect of Magnetic Field and Aggregation on Electrical Characteristics of Magnetically Responsive Suspensions for Novel Hybrid Liquid Capacitor

**Kunio Shimada** 

Faculty of Symbiotic Systems Sciences, Fukushima University, 1 Kanayagawa, Fukushima 960-1296, Japan; shimadakun@sss.fukushima-u.ac.jp; Tel.: +81-24-548-5214

**Abstract:** Magnetically responsive fluid based on polymers of natural rubber (NR-latex) involves a magnetic compound fluid (MCF) rubber liquid. For a wide range of engineering applications of suspensions or liquids with particles, their electrical characteristics of fluidic suspensions are investigated to obtain useful results that might be important in the study of devices, such as fluidic sensors and capacitors. The author of the present paper proposes that MCF rubber liquid can be produced by combining MCF and rubber latex. The influence of the aggregation of magnetic particles and rubber molecules on electrical characteristics under a magnetic field was investigated by measuring electrical properties under an applied voltage. The electrical characteristics change with a linear or a nonlinear response, based on conditions of particle aggregation. The capacity of the electric charge also changes with the conditions of particle aggregation. These results show that MCF rubber liquid is a novel hybrid capacitor.

**Keywords:** magnetic compound fluid (MCF); magnetic fluid; intelligent fluid; smart material; natural rubber; polymer; liquid; aggregation; particle; magnetic field; electrical characteristics; electric charge; capacitor; sensor; resistivity; electric conductivity; tunnel theory

## 1. Introduction

The utilization of the behavior and structure of particles dispersed in a polymer or colloidal suspension to enhance and control electric conductivity by magnetic and electric fields has attracted increasing attention. This property is required in many engineering applications such as sensors, capacitors [1], polishing [2], and conductive adhesives. For example, carbon nanomaterials are effective as magnetically responsive colloidal suspensions under magnetic and electric fields [3,4]. Fibrillary carbon nanotubes are often discussed [5] with contrasting electrorheological (ER) [6] and magnetorheological (MR) fluids [7] for many engineering applications such as those involving soft dampers and polishing liquids. Polymer-based composites have been implemented as smart materials [8] for flexible sensors [9]. Electrical properties are relevant to the applicability of conductive polymer composites for liquid sensing [10], insulators, conductive adhesives, and conductors [11,12]. In addition to the abovementioned suspensions, fluids, and composites, the utilization of ionic liquids to maintain magnetic response has also been reported [13].

The wide range of engineering applications for suspensions or liquids with particles indicates the need for study of the electrical or mechanical properties of fluidic suspensions or polymers under a magnetic or electric field for applications in the field of fluidic sensors. Sensors with polymer or colloidal suspension structures in a fluidic state have been investigated using various fabrication techniques: Liquid film utilizing water [14], composite molding liquid resin [15], composite polymer containing ionic liquid [16–19], and hybrid matrix doped by inorganic–organic composite materials [20,21]. These

sensing characteristics depend on the behavior of clusters of particles and molecules influenced by magnetic and electric fields. In other words, the aggregation of particles and molecules significantly affects the electric conductivity of the sensor. Our proposed sensor, which uses a magnetic compound fluid (MCF) as the magnetically responsive fluid, has the same features as these fluidic sensors [22–30].

The MCF contains nanometer-sized ordered magnetite ($Fe_3O_4$) particles, obtained using a magnetic fluid (MF), and micrometer-sized metal particles such as Fe and Ni [31,32]. The sensor consists of an MCF and diene-based rubber, such as natural rubber (NR), isoprene rubber, chloroprene rubber, butadiene rubber, nitrile rubber, or styrene–butadiene rubber, and is solidified by electrolytic polymerization through the application of a magnetic field [22,26]. Magnetic clusters of magnetic and metal particles [33], and rubber molecules structured by an applied magnetic field are fabricated heterostructures containing many thin-rod-shaped clusters that induce anisotropic and piezoelectric properties [25]. The state of the electrolytically polymerized MCF rubber sensor is solid. In contrast, the liquid state of MCF rubber, which was not electrolytically polymerized beforehand, is also sensitive. However, few characteristics of the fluidic MCF rubber sensor have been clarified. It may require the study of the effect of magnetic clusters on liquid-state MCF rubber as a liquid sensor. As for the behavior and structure of particles in liquid-state MCF rubber, the clarification of it aggregated by a magnetic field is needed. They may induce the variegated liquid sensor through the specific electric conductivity of rubber polymers, which contain particles as filler. The clarification is opposed to the investigation of particle aggregation in colloidal suspensions or fluidic composites [3–5,8,11]. In the first part of this study, we investigate the effects of aggregation on the electrical characteristics of MCF rubber liquid by evaluating the voltage and electric current applied to the inner liquid and external voltage.

In addition, as the liquid sensor senses the electric current and voltage generated through the application of a normal or shear force, the applied electric charge can also be stored in the inner part of the sensor if the liquid undergoes an oxidation–reduction reaction or if the electrodes have an electric double layer. Studies on the electric charge stored in suspensions or polymer liquids have been conducted on lithium ion batteries [34] in which particle dispersion causes aggregation of carbon [35–37], $TiO_2$ [38], and cobalt ferrite [39]. They often have an electrolyte base of polymer [40], ionic liquid [41,42], or ionic liquid crystal [43]. As MCF rubber can be considered an n-type or p-type semiconductor [28–30], electricity can also be used to charge the MCF rubber liquid so that it serves as a battery such as an electrolytic capacitor. In addition, the effect of the electric double layer on the MCF rubber liquid sensor should not be ignored. The MCF rubber liquid has the hybrid properties of a capacitor. The phenomena of electrically charging the MCF rubber liquid has not been clarified. Moreover, little is known about the effects of particle clusters in rubber-based polymers such as MCF rubber liquid. The purpose of the present study is to clarify the effect of aggregation on electric charge. Therefore, in the final section, the electric charge of the MCF rubber liquid is examined experimentally and discussed by evaluating the relationship between the measured output voltage and the applied input voltage.

## 2. Experimental Procedure

The experimental apparatus in Figure 1a was used to investigate the effect of particle aggregation due to the application of a magnetic field on the MCF rubber liquid. The detail composition of our used liquid will be shown in the following section. The electrodes were inserted into the MCF rubber liquid, which was poured into a nonmagnetic rectangular container, and a dc voltage was supplied between the electrodes. The liquid temperature was confirmed to be constant by measuring its temperature over the experiment. The electrodes were settled to a traversing device to adjust their height of the position by a screw and moved vertically. Then, the electric current and resistance of the liquid were measured by a voltmeter (NR-600, Keyence Co. Ltd., Osaka, Japan). The effects of electrophoresis and electric double layer on the wall of the inner side of the container were reduced by keeping the

electrodes away from the inner wall. These effects induce the electrification of the inner wall and, hence, must be eliminated from the measurement.

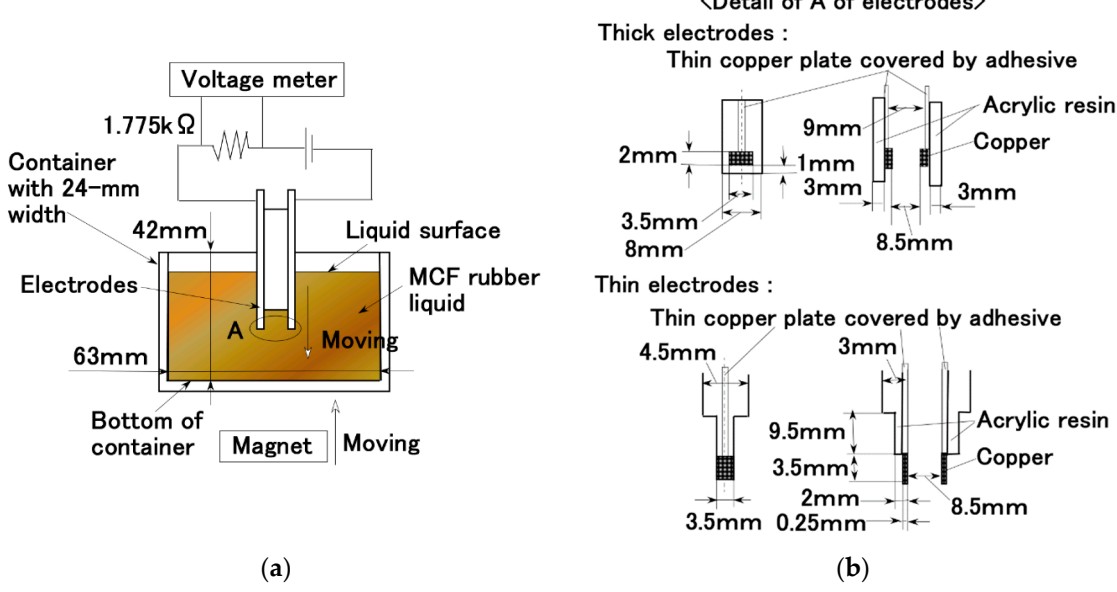

**Figure 1.** Schematic of experimental apparatus to investigate the effect of particle aggregation by the application of a magnetic field on the magnetic compound fluid (MCF) rubber liquid: (**a**) Overall diagram; (**b**) detailed diagram of electrodes denoted as A in (**a**).

We used a permanent magnet in order to investigate the effect of the magnetic field on the MCF rubber liquid. It is a neodymium magnet (Niroku Seisakusho Co., Ltd., Kobe, Japan) having $10 \times 15$ mm in size and 5 mm in thickness, and about 300 mT in magnetic field intensity of its surface. It was put on a plate settled to a traversing device to adjust its height of the position by screw. The positions of the magnet should be considered in the insertion of electrodes. Under dynamic conditions, the permanent magnet approaches the bottom of the container which has a 4-mm thickness at its bottom portion. On the other hand, under static conditions, the permanent magnet was installed underneath the container. Two magnet conditions were considered because of the sedimentation of the particles and molecules of the MCF rubber liquid, as shown in Figure 2a, under the static conditions of the MCF rubber liquid.

In Figure 2a, "A" denotes the area where NR-latex molecules are highly aggregated, "C" is the area with $Fe_3O_4$ and Ni particles, and "B" is the transition area between "A" and "C." We can confirm that "A" is part of dense NR-latex having dilute $Fe_3O_4$ and Ni, and "C" is one of dense $Fe_3O_4$ and Ni having dilute NR-latex because, originally, NR-latex has white color and $Fe_3O_4$ dark brown one. In the case without a magnet, when the electrodes are inserted into the MCF rubber liquid, the electric conductivity changes according to the location of the three areas shown in Figure 2b. In the case with a magnet installed underneath the container, shown in Figure 2c, the particles of the liquid aggregated excessively at the bottom of the container, making it difficult to insert the electrodes into the dense area of the aggregation; hence, the electrodes were inserted deeper into the dense area. The force applied to the inserted electrodes was measured by a strain gauge (KFG-5-120-C1-16, Kyowa Electronic Instruments Co., Ltd., Tokyo, Japan) installed on the electrodes. However, for the electrodes placed at the bottom of the container beforehand in Figure 2d, the closer the magnet moves to the outside bottom of the container, the more excessively the liquid particles aggregate. Because the electrodes were fixed to the same position from the beginning of the magnet's approach, the electrodes were buried in the dense aggregation. This way, the ease of electrode insertion varied according to the position of the magnet. Therefore, we used two cases of electrodes with different tip shapes, as shown in Figure 1b. For the cases in Figure 2b,c, thick electrodes squashed the aggregated particles, while

the thin electrodes easily penetrated the aggregated particles. In Figure 2d, the density of aggregated particles are different between in the types of thick and thin electrodes because of the existence of the nonmagnetic body of the acrylic resin adhered to the electrode metal of copper, as shown in Figure 1b.

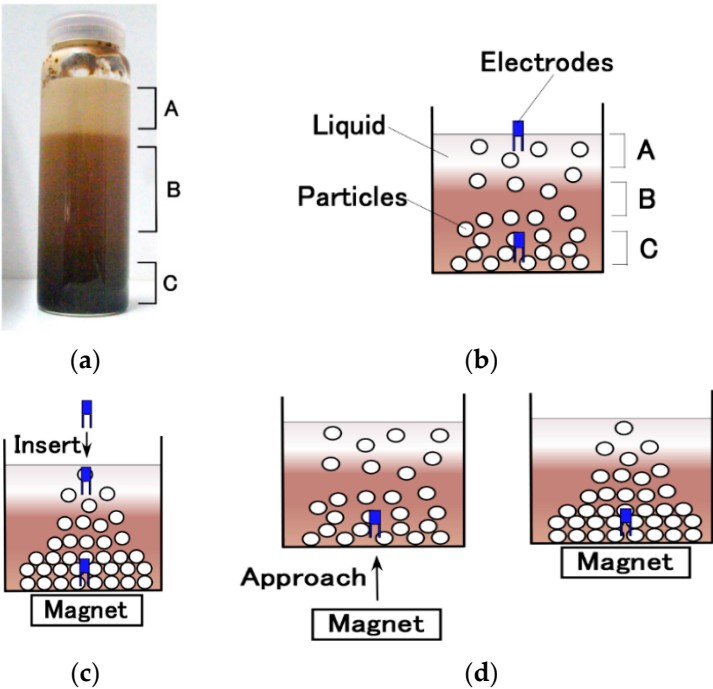

**(a)**          **(b)**

**(c)**          **(d)**

**Figure 2.** MCF rubber liquid in a rectangular container and the relation between the liquid and electrodes as shown in Figure 1 (**a**) Photograph of sedimentation of the MCF rubber liquid in the static condition; (**b**) the case without a magnet; (**c**) the case of a magnet installed underneath the container; (**d**) the case of electrodes placed at the bottom of the container beforehand and the magnet approaching the bottom of the container.

## 3. Electrical Properties of MCF Rubber Liquid

Another issue regarding the electrical properties of MCF rubber liquid needs to be clarified before investigating the effects of particle aggregation. We used the MCF rubber liquid and compared it with other liquids, as presented in Table 1. Ni is powder with particles on the micrometer order and pimples on the surface (No. 123 by Yamaishi Co. Ltd., Noda, Japan), MF is water-based with 50 wt % $Fe_3O_4$ (M-300, Sigma Hi-Chemical Co. Ltd., Tsutsujigasaki, Japan), and NR-latex from Rejitex Co. Ltd., Atsugi, Japan was used. Intelligent fluids such as MF, MR, and MCF have been developed and utilized in many engineering applications, such as those involving dampers, polishing, and sensors; thus, the presented data in this section are significant. In particular, the comparison of electric conductivity, as presented in Table 1, has rarely been confirmed. Therefore, these results are expected to play an important role. The relationship between the voltage and electric current of the liquid in the case without a magnet and with thick electrodes in "C" from Figure 2b, is shown in Figure 3. The arrows in the figure are the traces of electric current changed by the application of dc voltage. A uniform liquid shows a linear relationship between electric current and voltage of the liquid. Its gradient denotes the electrical resistance, which is presented in Table 1 and calculated from Figure 3. However, when the liquid involves NR-latex, the relation is nonlinear in the low voltage range and linear in other ranges. The electrical resistance presented in Table 1 is in the linear relation range in Figure 3. The linear and nonlinear relations are referred to as electron transfer via ohmic contact and Schottky contact, respectively. The former corresponds to a small potential barrier between the particles and molecules of the liquid, while the latter corresponds to a large potential barrier. In particular, NR-latex contains nonconductive molecules or particles. The behavior of electron transfer in the latter case depends on

the tunnel theory presented in a previous study on MCF rubber [28]. From the theoretical investigation using this theory, there is a relationship between the applied voltage $V_0$ and transmitted probability $T$, which is shown in Figure 4, where $T$ means electric current, and $b$ is the thickness of nonconductive rubber among the metal and $Fe_3O_4$ particles. $V_0$ and $b$ can initially be given as any value. These given values are within the appropriate range of the present experimental condition. Consequently, for a liquid with nonconductive molecules or NR-latex particles, a nonlinear relationship is exhibited between the voltage and electric current because of the lack of electric conductivity of the molecules or particles.

**Table 1.** Composition of liquids conducted on in the present experiment. MF: Magnetic field; NR: Natural rubber.

| | Ni [g] | MF [g] | NR-Latex [g] | Water [g] | Electric Conductivity [S/m × 10⁻²] |
|---|---|---|---|---|---|
| Water | | | | | 1.69 |
| MF | | 1.5 | | 15.5 | 11.3 |
| MCF | 6 | 1.5 | | 15.5 | 14.8 |
| NR-latex | | | 8 | 15.5 | 25.7 |
| NR-latex, Ni | 6 | | 8 | 15.5 | 23.5 |
| NR-latex, MF | | 1.5 | 5 | 15.5 | 25.5 |
| NR-latex, MCF | 6 | 1.5 | 8 | 15.5 | 29.8 |

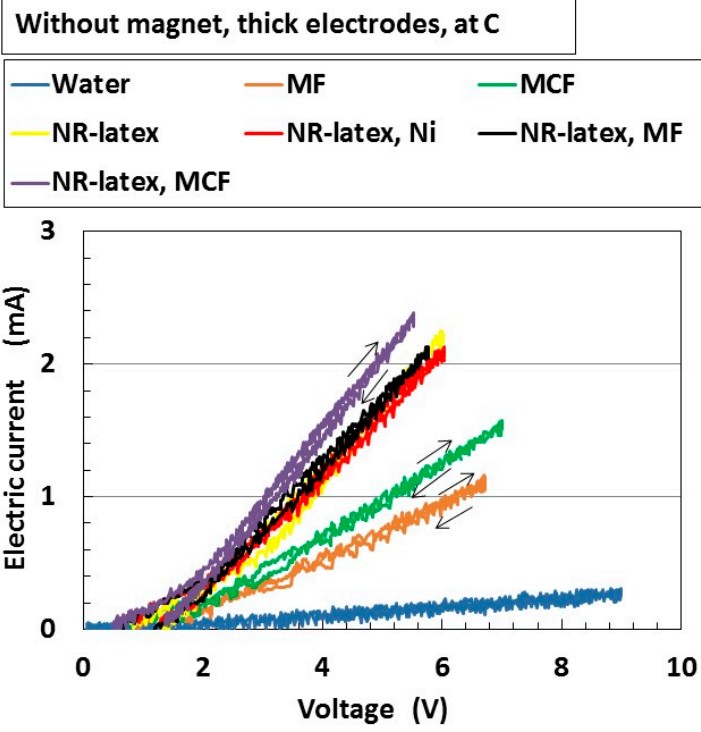

**Figure 3.** Relation between dc voltage and electric current of liquid in the case without a magnet and using thick electrodes in "C" in Figure 2b.

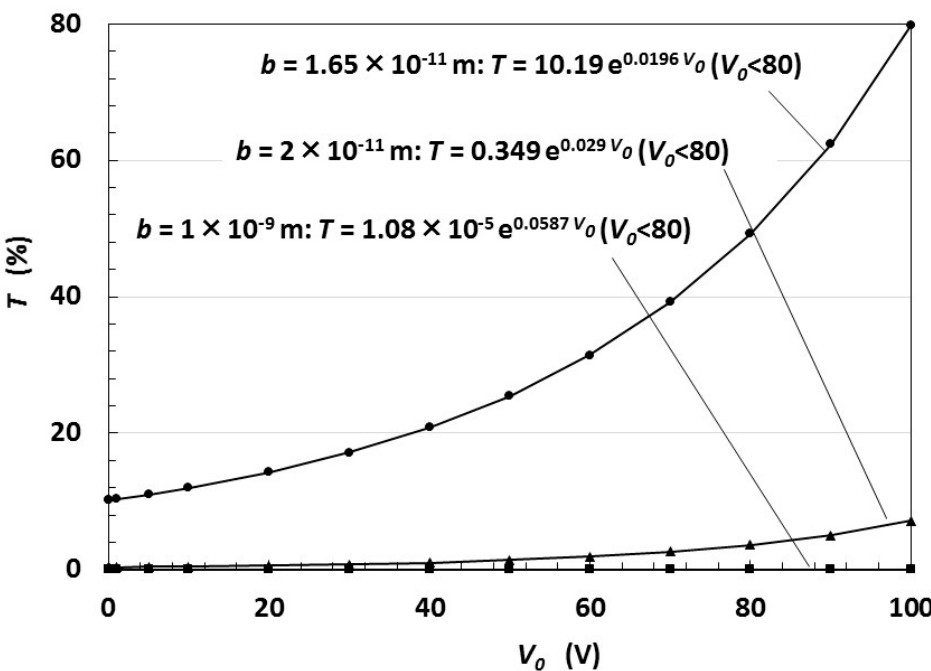

**Figure 4.** Theoretical results of the transmitted probability $T$ with respect to applied external voltage $V_o$ in relation to the thickness $b$ of nonconductive rubber between the metal or $Fe_3O_4$ particles.

We shall now look more closely at the results in Figure 3. For "NR-latex, MCF," Figure 5 shows resistivity in areas "A," "B," and "C," as shown in Figure 2a, by comparing the conditions of uniform dispersion of the MCF rubber liquid and by changing the components of the liquid. The data measured by using a voltmeter were confirmed to calibrate by commercially calibrated resistivity-measuring instruments (DS-72, Horiba Co. Ltd., Kyoto, Japan) in order to make them firm correct data with small error. Changes in each component were observed in the cases without a magnet and with thick electrodes. The resistivity decreased as the density of the sedimentation of particles and molecules of the MCF rubber liquid increased. As Ni, $Fe_3O_4$ particles, or NR-latex molecules increased, the resistivity decreased. These results are attributed to the large amount of electric current passing through the liquid with more conductive particles and molecules. The case of uniform dispersion is shown in areas "B" and "C."

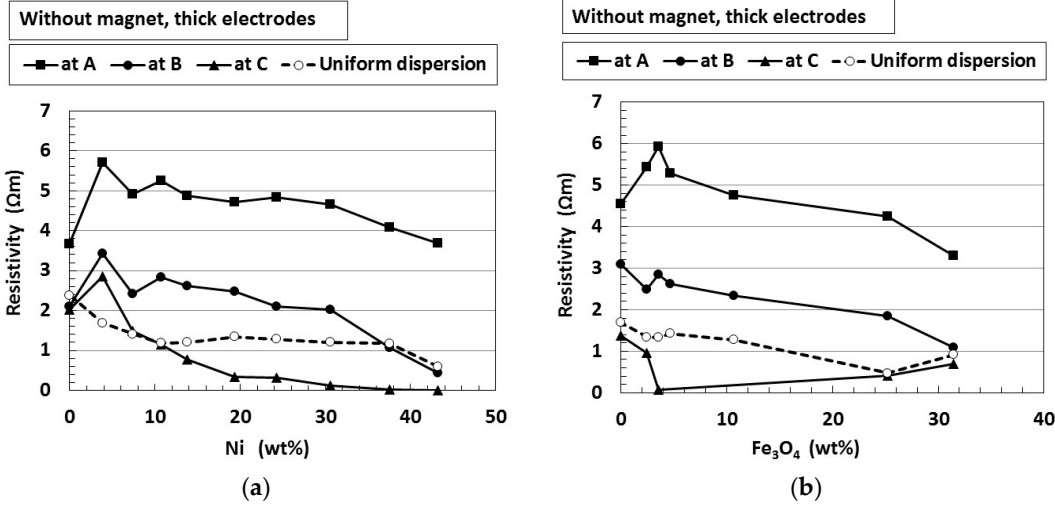

**Figure 5.** *Cont.*

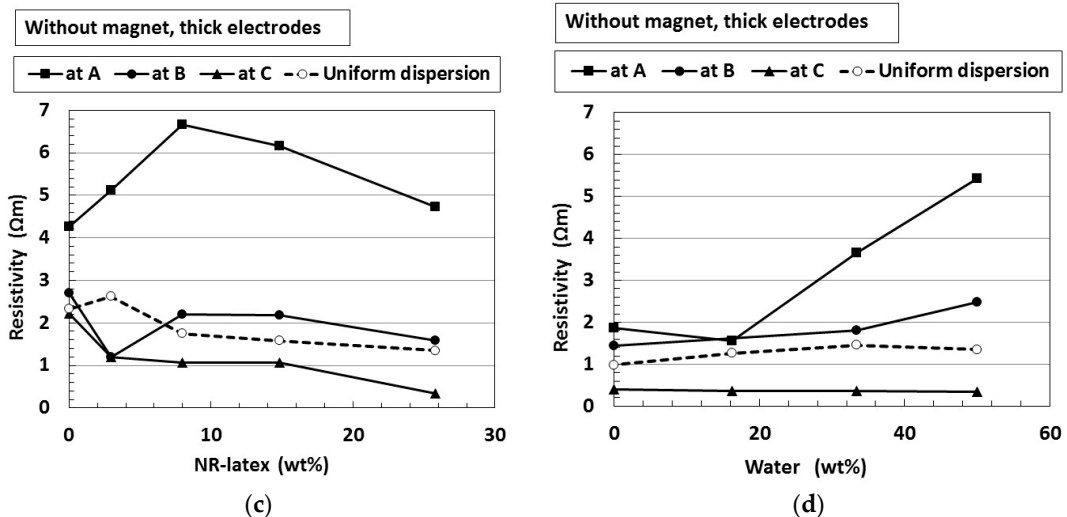

**Figure 5.** Resistivity to changes in each component of MCF rubber liquid in the case without a magnet and using thick electrodes in "A", "B", and "C" in Figure 2b and uniform dispersion: (**a**) Ni; (**b**) $Fe_3O_4$; (**c**) NR-latex; (**d**) water.

## 4. Effect of Particles Aggregation

We now consider the effect of applying a magnetic field on particle aggregation. The liquid used in this section is "NR-latex, MCF," presented in Table 1. First, we investigate the case of moving electrodes in Figure 2b,c. Figure 6 uses thin electrodes, and the position of a magnet is underneath the container, as shown in Figure 2c. In the abscissa, from right to left, the electrodes sweep for measurement. Because the left end of the abscissa indicates the bottom of the container, at this position, the magnetic field strength is the highest. Therefore, more Ni and $Fe_3O_4$ particles aggregate toward the left side of the abscissa, which increases the electric conductivity, electric current, and force applied to the inserted electrodes while the voltage in the liquid decreases. As more the particles aggregate, as more the current flows, however, the voltage decreases. Therefore, there is correlation between the current and the voltage, because the current can flow more easily inside the denser particles aggregation. The role of the magnet's motion is to create the aggregation. The nearer the magnet approaches to the container, the more the particles aggregate.

From Figure 6, at an electrode position at the bottom of the container, the relationship between the voltage and electric current of the MCF rubber liquid is shown in Figure 7. The arrows indicate the traces of electric current changed by the application of voltage. Near the bottom of the container, where the magnetic field strength is greater and the particles are more aggregated, the relationship is linear. However, the particle aggregation decreases with distance from the bottom of the container in a nonlinear manner. These results are attributed to the larger gap between Ni and $Fe_3O_4$ particles. When the position of the electrodes is farther away from the bottom of the container, electron transfer becomes difficult and is dominated by the tunnel effect.

Figure 8 shows a comparison of the changes in electric conductivity with the distance between thin and thick electrodes. The electric conductivity of thin electrodes is larger than that of thick electrodes. This is because the thick electrodes smash the aggregated particles, while the thin electrodes are inserted easily into the aggregates.

Next, we investigate the case of a moving magnet with electrodes settled at the bottom of the container. Figure 9 shows the voltage, electric current, and electric conductivity of the MCF rubber liquid, and force applied to the electrodes. Changes in magnetic field strength when the magnet approaches the bottom of the container indicates that the magnetic field strength increases as the magnet approaches. As the magnet approaches the bottom of the container, the particles aggregate to the bottom of the container, which increases the electric current flows and electric conductivity but

decreases the voltage. In addition, the aggregation becomes denser, which increases the force applied to the electrodes.

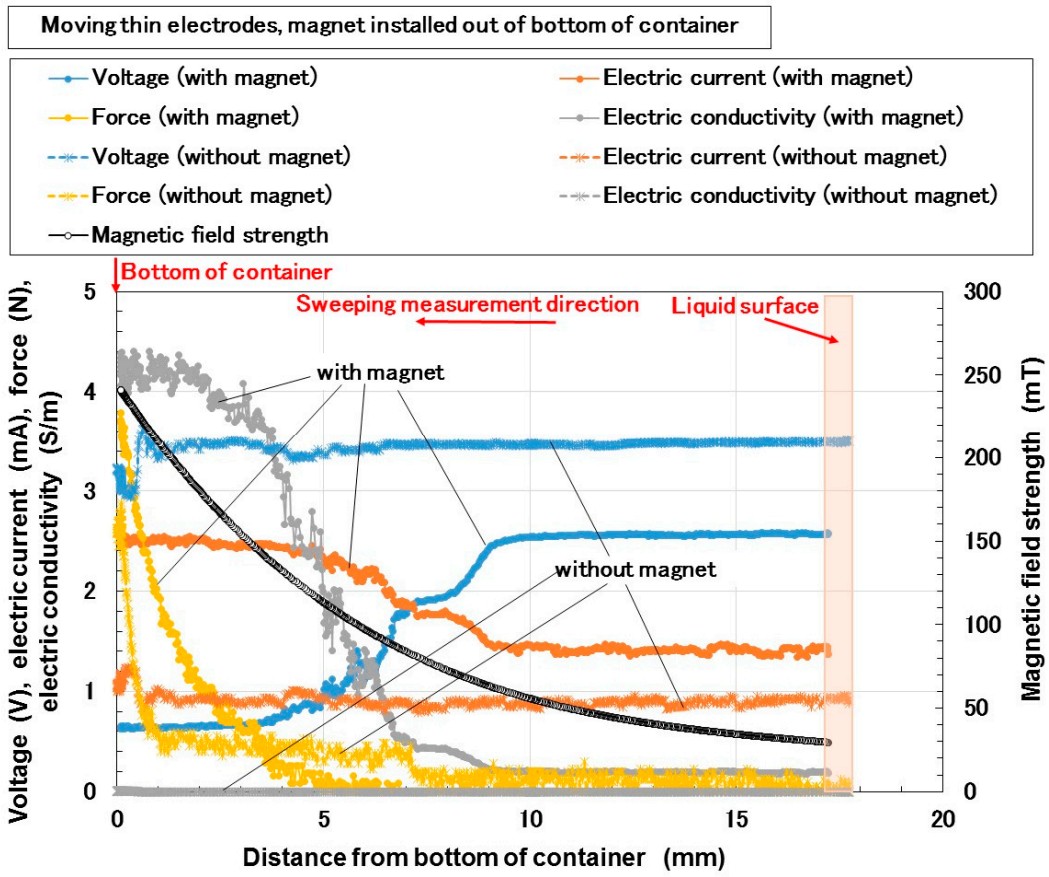

**Figure 6.** Changes in voltage, electric current, and electric conductivity of MCF rubber liquid, force applied, and magnetic field strength with distance from the bottom of container for moving electrodes with and without a magnet, with reference to Figure 2b,c.

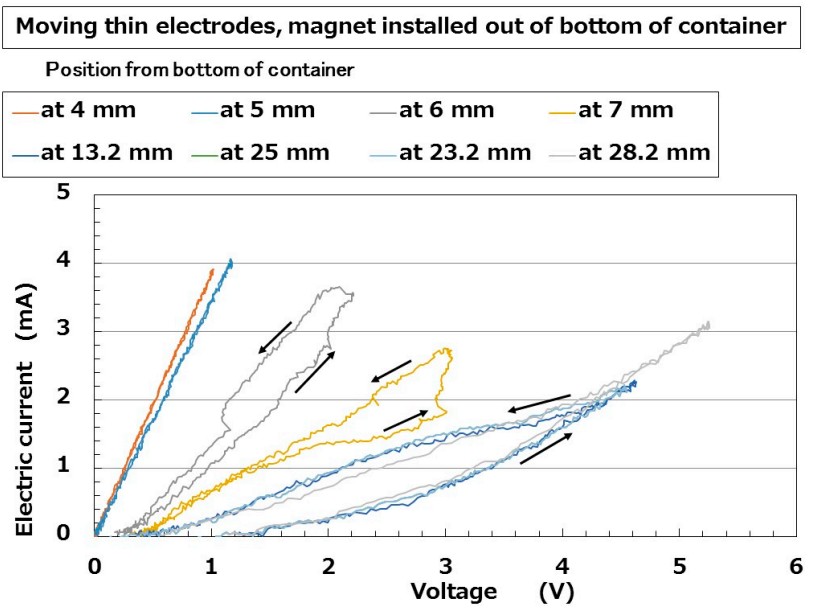

**Figure 7.** Relation between voltage and electric current of MCF rubber liquid for moving electrodes with installation of a magnet at the bottom of the outer wall of the container regarding Figure 6.

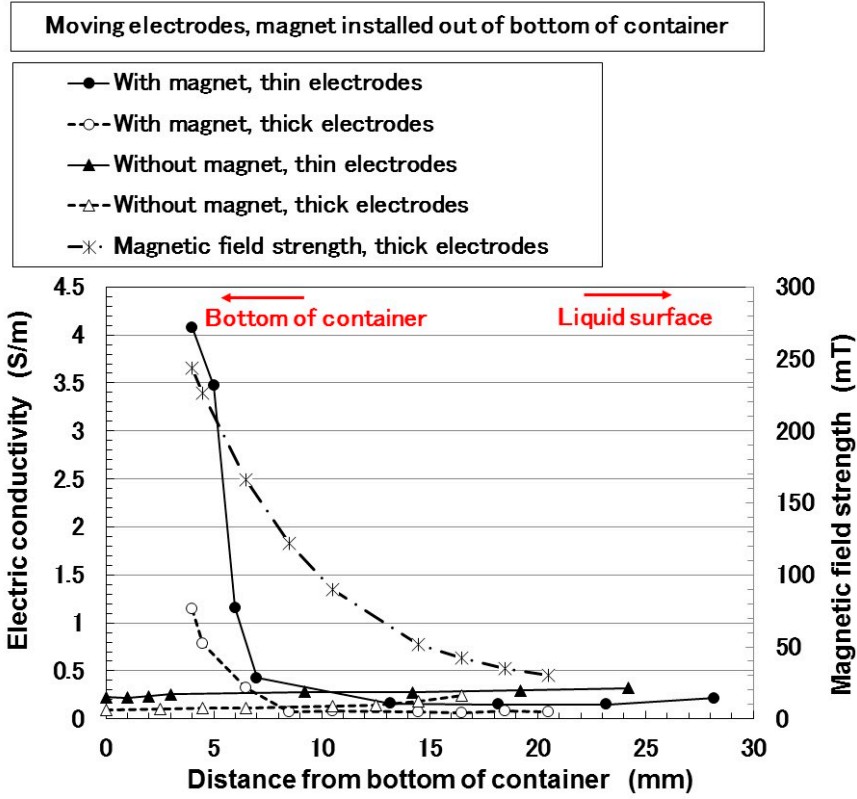

**Figure 8.** Comparison of electric conductivity with changes in the distance of the container between thin and thick electrodes for moving electrodes with installation of a magnet at the bottom of the outer wall of the container.

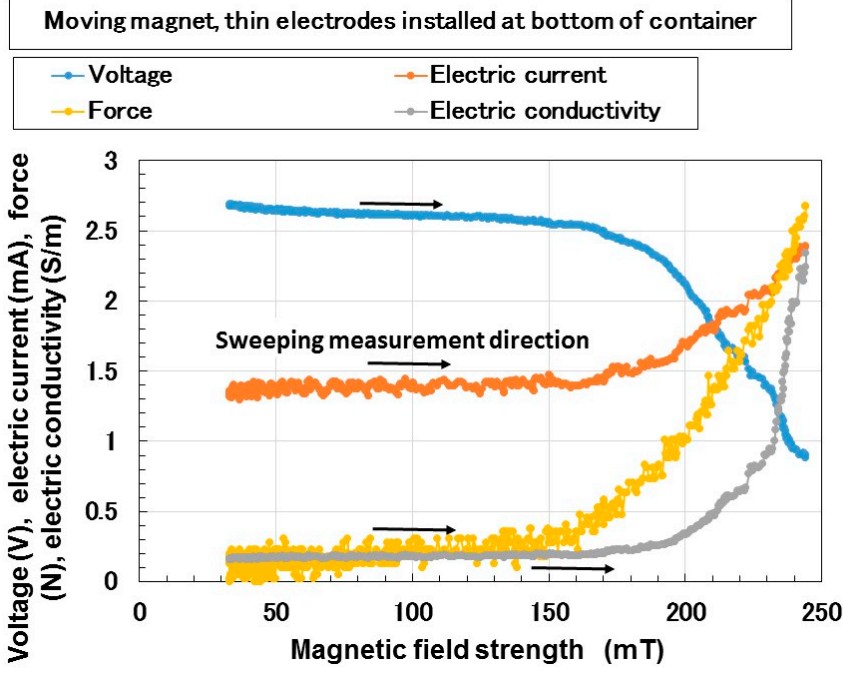

**Figure 9.** Voltage, electric current, and electric conductivity of MCF rubber liquid, and force acting on the electrodes for a moving magnet and with electrodes installed at the bottom of the container.

Figure 10 shows the comparison of electric conductivity between the thin and thick electrodes. Just as the results in Figure 8, the electric conductivity of thin electrodes is larger than that of thick electrodes.

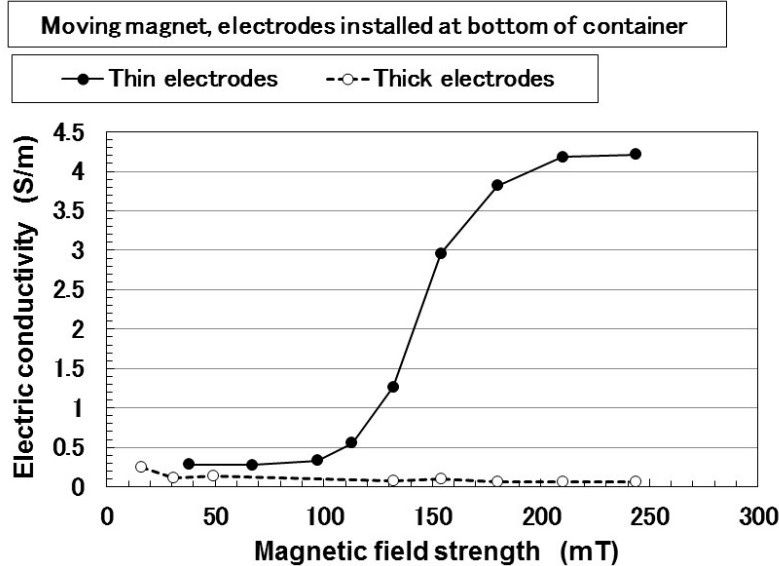

**Figure 10.** Comparison of electric conductivity between thin and thick electrodes for a moving magnet and with electrodes installed at the bottom of the container.

Figure 11 shows the relation between the voltage and electric current of the MCF rubber liquid at some distance for the magnet from the bottom of the container. The arrows in the figure are the traces of the electric current adjusted by changes in the voltage. As the magnetic field strength increases, the particles become increasingly aggregated because the magnet is closer to the bottom of the container, and the relation is linear.

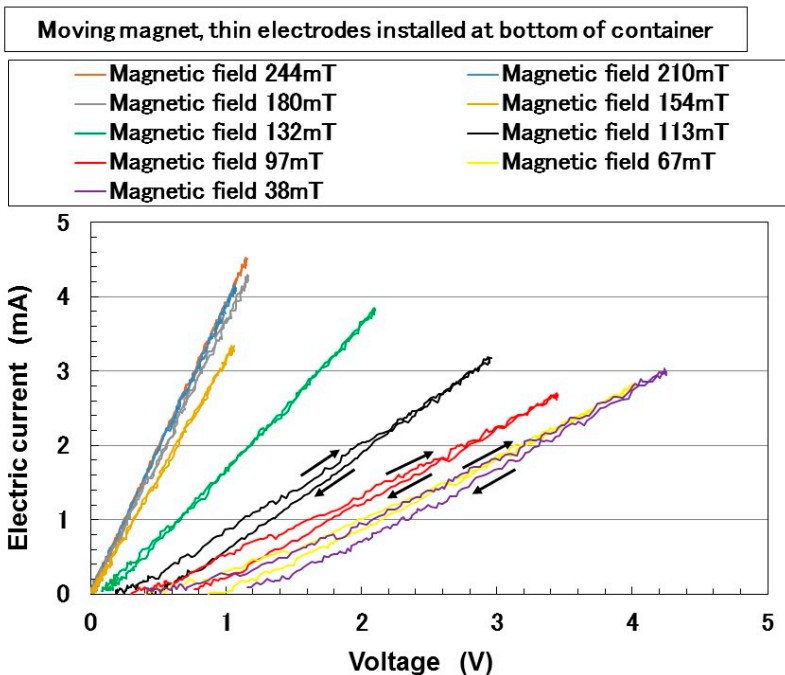

**Figure 11.** Relation between voltage and electric current of MCF rubber liquid at a position of the magnet from the bottom of the container for a moving magnet and with electrodes installed at the bottom of the container.

Finally, we need to ensure that the experimental results in this section as well as in Section 3 on liquids involving NR-latex also include the effect of the solidification of MCF rubber by electrolytic polymerization. As seen from previous studies [22–30], the MCF rubber liquid begins to solidify the moment an electric field is applied. Therefore, in the experiment, the solidification of the MCF rubber liquid can be considered to occur around the electrodes. However, the solidification effect is not excluded from the experimental data. Solidification has the following effects [22–25]. The electric conductivity of NR-latex with MCF is larger than that without MCF. The electric conductivity with the application of a magnetic field is larger than that without a magnetic field, and it increases with magnetic field strength. These solidification effects must be considered in the present experimental data. However, the present experiment was conducted only by replacing the electrodes each time a voltage was applied to the MCF rubber liquid. This was because the solidification effect should be avoided as much as possible.

## 5. Electric Charge

The experimental procedure detailed in Section 4, which involved the consideration of the electric current or voltage of the MCF rubber liquid in the case of application of an external dc voltage, can be equated with another procedure in which the restored voltage in the MCF rubber liquid (we call it "output voltage" for convenience) was measured in the case of the application of an external dc voltage (we call it "input voltage" for convenience). Specifically, this corresponds to the equivalent experimental apparatus shown in Figure 12. This means that MCF rubber is also a capacitor. This can be expressed theoretically using the same tunnel theory applied in a previous study [30], as shown in Figure 4. The dimensionless capacitance $C^*$ is calculated by Equation (3) in reference [30], where $C^* = CV/Q$, $V = n\Delta V_j$, $Q = en_1$, $n$ is the number of segments of the metal particles, $e$ is the charge of the electron, and $n_1$ is the charge on the surface of the MCF rubber. The dimensionless capacitance $C^*$ of the MCF rubber liquid is presented as the relation between the applied voltage $V_0$ and $C^*$, where $b$ is the same parameter as in Figure 4, shown in Figure 13. Therefore, in the present study, we will experimentally clarify the relation between the output voltage and input dc voltage, using the experimental apparatus illustrated in Figure 12.

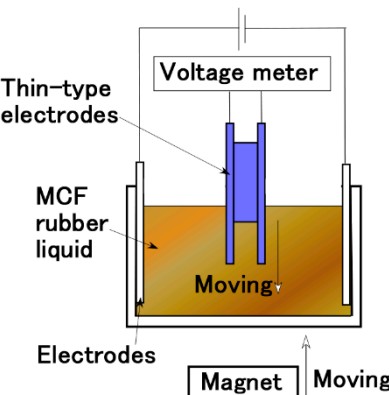

**Figure 12.** Schematic diagram of experimental apparatus to investigate the output voltage through MCF rubber liquid to input of an external dc voltage.

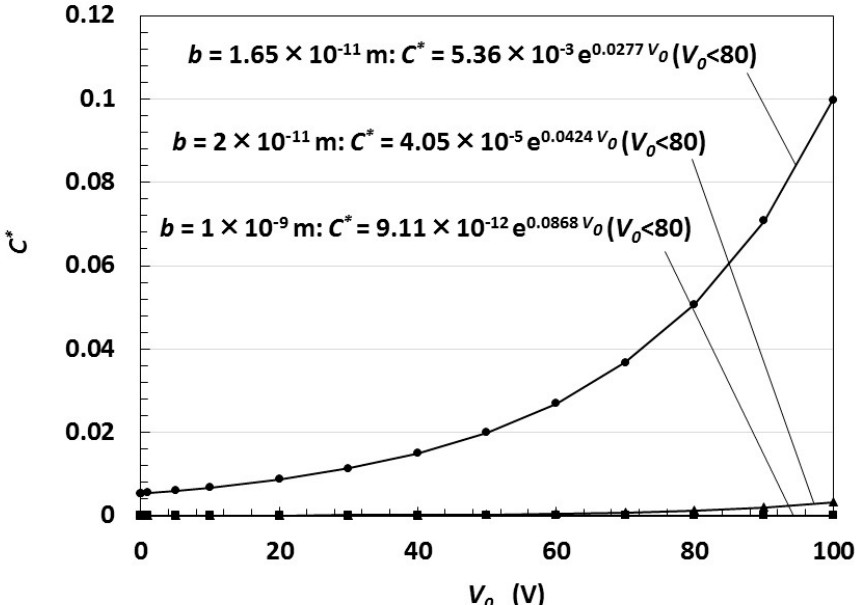

**Figure 13.** Theoretical results of the dimensionless capacitance *C** to applied external voltage $V_o$ in relation to the thickness *b* of the nonconductive rubber between the metal or $Fe_3O_4$ particles.

As seen in Figure 7; Figure 11, with the application of a magnetic field, the relationship between the voltage and electric current of the MCF rubber liquid can change from linear to nonlinear according to the particle aggregation. Therefore, the MCF rubber liquid is feasible for both rechargeable batteries and electric double-layer capacitors, and the difference can be realized by applying a magnetic field. The former has a nonlinear relationship, and the charging and discharging times are comparatively long, which corresponds to a battery as well as a polymer electrolytic capacitor. It is also dependent on the chemical response around the electrodes. However, the latter has a linear relationship between the electrical charge and voltage, and the charging and discharging times are short, which corresponds to a double-layer capacitor. It is also dependent on behavior of ionized molecules or particles charged based on electric double-layer. The MCF rubber liquid exhibits the behaviors of both ionized molecules and particles as well as the corresponding chemical responses. Therefore, the MCF rubber liquid can also be feasible for both electric double-layer capacitors and polymer electrolytic capacitors. In the present study, we will clarify the capability of the electric charge. Regarding the electric double-layer capacitor, there has been a study on the electrochemical double-layer capacitor using carbon and ionic liquid [44]. Our study indicates a novel capacitor that utilizes NR-latex.

We evaluated the electric charge of the MCF rubber liquid by measuring the output voltage and input dc voltage in the experimental apparatus, as shown in Figure 12. Using the same procedure as in Section 4, the electrodes or magnet was moving and thin electrodes were used. The liquid used in the present section is "NR-latex, MCF", presented in Table 1.

Figure 14 shows the input dc voltage and output voltage in the case of the magnet approaching the bottom of container and thin electrodes installed at the bottom of container. The output voltage to input voltage as shown in every following figure in this section involving Figure 14 connotes the capacity of the electric charge. The magnetic field intensity presented in the figure is at one of the electrode positions. The relation between the input and output voltage is linear. This tendency also appears in the MCF rubber liquid previously presented in Sections 3 and 4 in the following cases: Case without electrodes at larger voltage, as seen in Figure 3; case of moving electrodes closer to the bottom of container, which leads to greater dense aggregation, as seen in Figure 7; and case of moving magnet at larger magnetic field strength at the bottom of container, which leads to greater dense aggregation, as seen in Figure 11. These indicate that dense aggregation influences the charging behavior of ionized molecules or particles based on the electrical double-layer. This tendency is also shown in Figure 14. If

the MCF rubber liquid has more diluted aggregation, then the relation between the input and output voltage could be nonlinear.

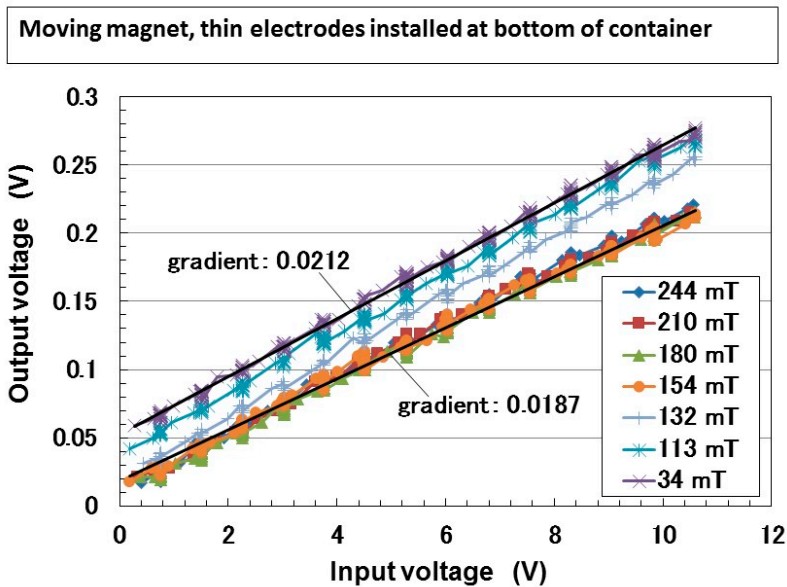

**Figure 14.** Relation between input and output voltage at a position of the magnet from the bottom of the container for a moving magnet and with electrodes installed at the bottom of the container.

The gradient of output to input voltage, shown in Figure 14, decreases with increasing magnetic field strength; hence, the capacity of the electric charge decreases as the aggregation density increases. Where the uncertainty of these data denotes the coefficient of determination on the linear regression, which presents as value within 0–1, and the reliability becomes greater as closer to 1. The "gradient: 0.0212" has the coefficient of determination 0.998 and the "gradient: 0.0187" the coefficient 0.9957. Therefore, these values have sufficient reliability. To clearly explain this tendency, the relation of output voltage to magnetic field strength is shown in Figure 15. This tendency occurs for aggregation created at the electrode positions, and it is different from the case of aggregation created beforehand, as shown in the following figures.

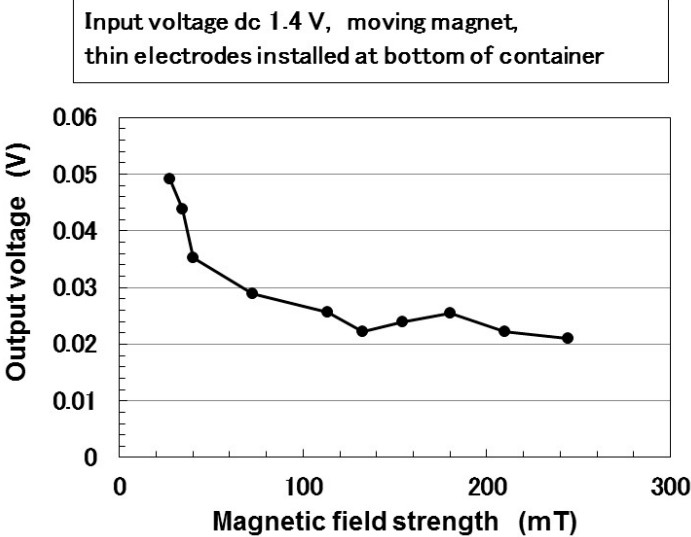

**Figure 15.** Relation of output voltage and magnetic field strength at a position of magnet from the bottom of container for a moving magnet with electrodes installed at the bottom of the container and application of an input voltage of dc 1.4 V.

Figure 16; Figure 17 show the output voltage of moving electrodes without and with a magnet at the bottom of the outer wall of the container, respectively. In these cases, aggregation was created beforehand, which was then penetrated by the electrodes. In the area of dense aggregation, the capacity of the electric charge increases regardless whether the dense aggregation was created by sedimentation without a magnet or by agglomeration with a magnet. Figure 14,Figure 15,Figure 16,Figure 17 show that the capacity of the electric charge varies with the position of the electrodes inside or outside the aggregation.

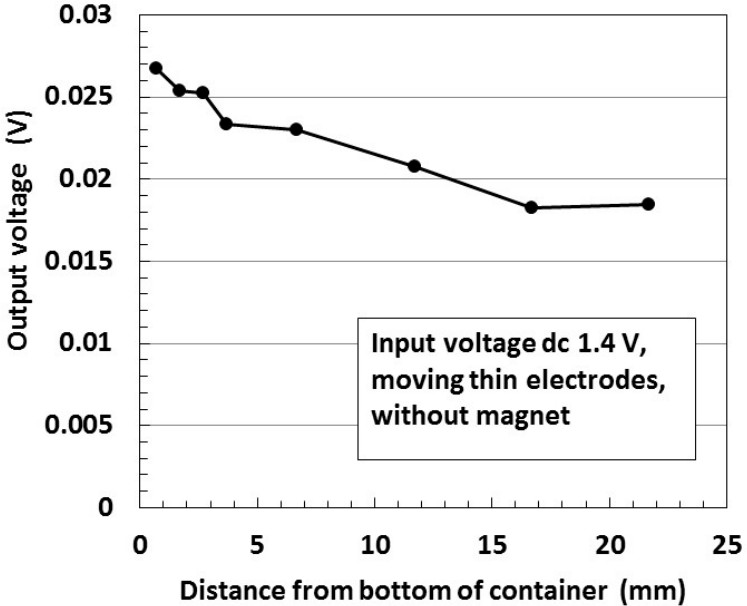

**Figure 16.** Output voltage and the distance of container for moving electrodes without a magnet and the application of dc 1.4 V of input voltage.

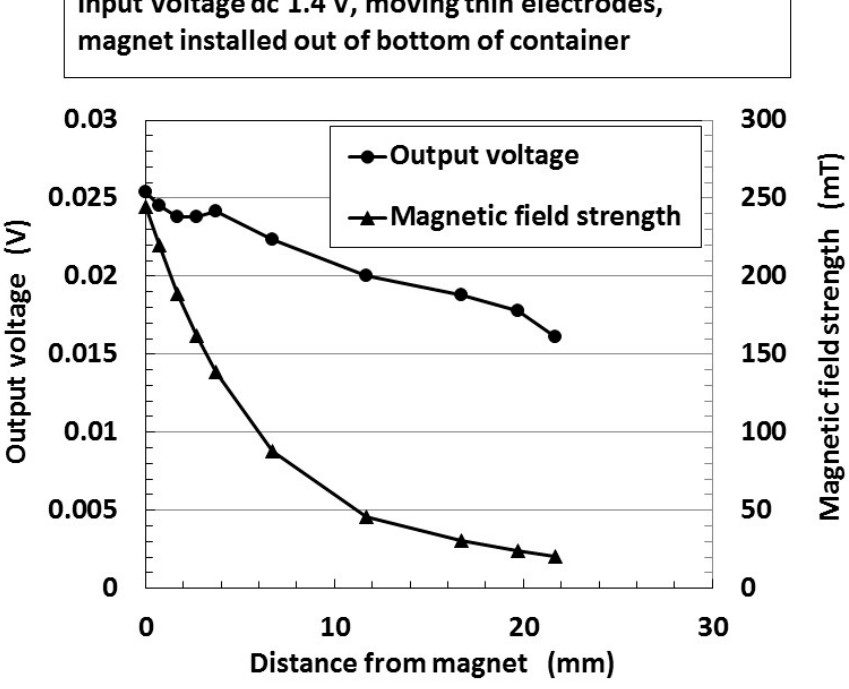

**Figure 17.** Output voltage and distance of container for moving electrodes with installation of a magnet at the bottom of the outer wall of the container and the application of dc 1.4 V of input voltage.

## 6. Conclusions

We clarified the effects of the magnetic field and aggregation on the conductivity of the magnetic responsive fluid, MCF rubber liquid, as a novel hybrid liquid that serves as a capacitor.

The relation between the voltage and electric current of the MCF rubber liquid changes from linear to nonlinear according to the particle aggregation with the application of a magnetic field. At higher voltage or magnetic field strength, the relation between the voltage and electric current becomes linear as the aggregation density increases. Therefore, the MCF rubber liquid is feasible for both rechargeable batteries and electrical double-layer capacitors, and the difference can be realized by the applied magnetic field strength.

However, with respect to aggregation at the position of the electrodes, with the application of a magnet, the capacity of the electric charge decreases as the aggregation density increase. This tendency differs from the case of aggregation created beforehand and the electrodes inserted into the aggregation. In the area of denser aggregation, the capacity of the electric charge increases regardless of whether the dense aggregation was created by sedimentation without a magnet or by agglomeration with a magnet. Thus, according to the position of electrodes inside or outside the aggregation, the capacity of the electric charge is different. We must consider the correlation position between the electrodes and magnetic field when designing a capacitor or battery using MCF rubber liquid.

**Author Contributions:** For this research article, K.S. conceived, designed, and performed the experiments; analyzed the data; and wrote the entire paper.

**Funding:** This work was supported in part by JSPS KAKENHI Grant Number JP 18K04040.

**Conflicts of Interest:** The founding sponsors had no role in the design of the study. The author declare no conflict of interest.

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
