# Peer review of "Effect of Magnetic Field and Aggregation on Electrical Characteristics of Magnetically Responsive Suspensions for Novel Hybrid Liquid Capacitor"

_magnetochemistry, doi:10.3390/magnetochemistry5020038_

Round 1
Reviewer 1 Report
The paper presents a study of the electrical characteristics of magnetically responsive suspensions due to aggregation in the presence of a magnetic field. Such magnetic suspensions are of interest for the design of novel capacitors. The title of the paper is generally appropriate but should be revised grammar-wise. The abstract summarizes well the approach and the main findings and covers the main aspects of the work. The paper is well organized and comprehensively written. In the introduction, the work is put into proper context to previously published research. The experimental procedure is explained well, but some details should be added, see my comments 2 and 3. The results are discussed comprehensively in much detail and are well understandable. The figures and illustrations are clear and appropriate. In the conclusion, the experimental results are well summarized, with an emphasis on envisioned future capacitor or battery applications.
I have some questions and some suggestions for improvement:
1) What is the difference between capacitor and condenser? I looked it up at quora.com and found “Condenser refers to different objects in different engineering fields. When electronic circuits are considered, condenser means a capacitor. In thermodynamics, condenser is a device which condenses (converts into liquid) gaseous materials by cooling. In optics, condenser is a device that helps concentrate light. Among these different uses of the word, thermodynamic term is the most common.” Please explain what you mean by condenser. Do you mean a condenser for solidification of the rubber liquid?
2) Section 2: In line 89, it says: “The position of the magnet ..”. I was surprised when reading this because there was no mention of a magnet previously. Please explain which type of magnet you used (material, size, minimum/maximum distance to liquid in container ..).
3) Section 2 and 3: I suggest to mention the manufacturers and types of the voltmeter and the “commercially calibrated resistivity-measuring instruments” (line 233) you used, also the strain gauge (line 147), so that the reader gets an impression of the accuracy of the measurements. How was the moving of magnet and electrodes (section 5, Fig. 12) done? By hand, by a screw, or by a motorized translation stage?
In addition, I noticed the following grammar and punctuation typos:
Title: better: “.. on electrical characteristics of magnetically responsive suspensions ..”
P. 1, lines 9-13: this sentence is very long. I had a hard time to understand it. I suggest to split it into shorter sentences.
P. 1, line 17: “.. nonlinear response, based on ..”
P. 2, line 48: don’t you mean “features” instead of “feasibility” ?
P. 2, line 52: second part of sentence has no verb. Better: “ .. rubber, and is solidified ..”
P. 2, lines 59-63: this sentence is very long. I don’t really understand it. Please re-phrase.
P. 2, line 63: better: “In the first part of this study, ..”
P. 3, line 158: better: “3. Electrical properties of MCF rubber liquid”
P. 3, line 160: better: “We used MCF rubber liquid ..” because you don’t mean a specific one, do you?
Author Response
I appreciate for your valuable comments and suggestions for the present our report. According to their comments, we would like to reply for them as follows.
Comment 1: As you pointed the difference between capacitor and condenser, they must be unified or corrected. As for the present article, capacitor is the most suitable. Therefore, they are corrected.
Comment 2: The detail explanation of our used magnet is added in line 90, according to your point.
Comment 3: According to the suggestions, we added and corrected the information of our used voltmeter and strain gauge in lines 86, 153, 243. In addition, we added the information of our used powders and liquids in line 168. As for our procedure of magnet and electrodes, the detail explanations are added in lines 85, 90.
Other comments:
/ According to the advice, the part of the title is corrected to “Electrical Characteristics of Magnetically Responsive”.
/ According to the advice, the sentences in line 9 are corrected.
/ According to the advice, the sentences in line 17 are corrected.
/ According to the advice, the word in line 48 is corrected.
/ According to the advice, the word in line 51 is corrected.
/ According to the advice, the sentences in line 58 are corrected.
/ According to the advice, the word in line 63 is corrected.
/ According to the advice, the sub title in line 167 is corrected.
/ According to the advice, the sentence in line 163 is corrected.

Reviewer 2 Report
The idea of the manuscript may be interesting, but the author fails iin making this clear to the reader. This is my main concern, but others follow:
In the Introduction it is very important to comment on the objectives of the investigation, the advances and improvements that it means, a clear statement of the main asppects of the paper,...
Line 49: what does the author mean by "ordered magnetite"?
L 75: what is the difference between capacitor and condenser?
L 68: How can we "store" voltage?
L. 81: I miss details on the fluid preparation, composition, sources, instrumental tools,...
L. 86: no heating of the fluid was observed upon application of the voltage?
L. 140: the author claims that A is highly aggregated, but I see a clear supernantant. Please explain
L. 178: T transmitted probability and also current?
Table 1.: The resistance is meaningless if one does not know the geometry of the system. Conductivity is better
Figure 4; What is Vo? How can one determine b?
Figure 6: It appears as if both voltage and current were independent quantities, but they are not. Furthermore, I cannot see sufficient discussion on the large amount of data in Figure 6
What is the role of magnet motion? How is it performed?
L 509: How is the dimensionless capacitance calculated?
Fig. 14: gradient means slope? The uncertainty (+-) in the slopes is needed so as to infer if there is significant differences between data
L. 637; capacity of the electric charge?
Author Response
I appreciate for your valuable comments and suggestions for the present our article. According to their comments, we would like to reply for them as follows.
Comment 1: By another reviewer, the introduction is evaluated to be suitable for objectives, advances, and so on. According to your suggestion, we check to correct the introduction.
Comment 2: In line 49, “μm-sized metal ordered particles“ is misspelling to be corrected.
Comment 3: The same suggestion is pointed out by another reviewer. “Condenser” and “capacitor” should be corrected or unified. In the present article, “Capacitor” is suitable. In addition, the words in other sentences are to be corrected.
Comment 4: In general, the principle of capacitor’s storing is based on an oxidation‐reduction reaction or an electric double layer. To avoid confusion, the pointed sentence is corrected.
Comment 5: The preparation and composition of our used liquid, instrumental tools, and so on are added in the sections 2 and 3.
Comment 6: The liquid temperature was confirmed to be constant by measuring its temperature over the experiment, so the heating of the liquid did not exist. This comment is added in the article.
Comment 7: We can confirm that “A” in Fig 2(a) is part of dense NR-latex having dilute Fe3O4 and Ni, and “C” of dense Fe3O4 and Ni having dilute NR-latex, because originally NR-latex has white color and Fe3O4 dark brown one. In order to avoid the confusion, we added this comment.
Comment 8: As shown in line 194, transmitted probability T denotes electric current in the quantum mechanics field.
Comment 9: According to the suggestion, resistance in Table 1 is corrected to conductivity.
Comment 10: As presented in reference 28, Vo and b can initially be given as any value. These given values are inner the appropriate range of the present experimental condition. We added this comment.
Comment 11: In Fig. 6, as more the particles aggregate, as more the current flows, however, the voltage decreases. Therefore, there is correlation between the current and the voltage. Because the current can flow more easily inner the more particles aggregation. The role of the magnet’s motion is creation of the aggregation. As nearer the magnet approaches to the container, as more the particles aggregate. The clear explanation is lack, I think. Therefore, these are added.
Comment 12: Dimensionless capacitance C* is calculated Eq. (3) in Reference 30. The detail explanation is added.
Comment 13: The uncertainty may denote the coefficient of determination on the linear regression, which presents as value within 0 – 1, and the reliability becomes greater as closer to 1. The explanation is added.
Comment 14: The output voltage to input voltage connotes the capacity of the electric charge. This explanation is added.

Round 2
Reviewer 2 Report
The author has taken into consideration all my previous comments on this paper. Although some Englishr evision is still needed, from my side I consider the paper acceptable.